# Drowning in Children: Retrospective Analysis of Incident Characteristics, Predicting Parameters, and Long-Term Outcome

**DOI:** 10.3390/children7070070

**Published:** 2020-07-01

**Authors:** Liliane Raess, Anna Darms, Andreas Meyer-Heim

**Affiliations:** Swiss Children’s Rehab, University Children’s Hospital Zurich, 8910 Affoltern a. Albis, Switzerland; liliane.raess@kispi.uzh.ch (L.R.); anna.d@gmx.ch (A.D.)

**Keywords:** child, drowning, long-term outcome, rehabilitation, incident characteristics, prognostic parameters

## Abstract

Background: Drowning is the second leading cause of unnatural death in childhood worldwide. More than half of the drowned children, who were in need of cardiopulmonary resuscitation (CPR) at the scene suffered from lifelong neurological sequelae. There are few data about prognostic predictors in the pediatric population of drowning victims. The objective of the study was to assess incident characteristics, prognostic parameters, and long-term outcome of children recovering from a drowning incident. Methods: We carried out a retrospective analysis of data of the cohort of pediatric cases (age 0–18) of drowning victims admitted in the years 2000–2015 to the emergency room/intensive care unit/pediatric ward at the University Children’s Hospital of Zurich, Switzerland. Outcome was classified by the Pediatric Cerebral Performance Category Scale (PCPCS). New subcategories of severity for known prognostic parameters have been defined. A correlation analysis was performed between the subcategories of the prognostic parameters and the PCPCS. Results: A total of 80 patients were included in the analysis. Of these, 64% were male, most of the patients were at the age of 0–5 years. More than 80% of the patients were unattended at a public or private pool when the drowning incident happened. In all, 61% (*n* = 49) needed cardiopulmonary resuscitation (CPR). Of the resuscitated children, 63% showed good to mildly impaired long-term outcome (PCPCS 1–3). Furthermore, 15% (*n* = 12) were transferred to rehabilitation. Seven children died during the hospital stay and another four died due to complications in the ten years following the incident. The newly defined subcategories of the parameter submersion time, Glasgow Coma Scale (GCS) at time of admission, body temperature at time of admission, blood pH, blood glucose, and blood lactate level correlated significantly with the PCPCS. Conclusions: Supervision of children, especially boys of the age 0–5 years, next to public or private pools is most important for prevention of drowning incidents in Switzerland. Cardiopulmonary resuscitation done by trained staff leads to a better long-term outcome. Medical decision making in severe cases of drowning should consider submersion time, GCS at time of admission, body temperature at time of admission, blood pH, blood glucose, and blood lactate levels, as these parameters correlate with long-term outcome.

## 1. Introduction

Drowning is considered the second most frequent cause of preventable death in children worldwide after traffic incidents and is the most frequent cause of cardiovascular arrest in children and adolescents [1,2,3,4]. In Switzerland, an average of fourteen children drown fatally each year. In addition, each year there are about 270 non-fatal drowning (children and adults) incidents. Numerous definitions of “drowning” exist, which were standardized in 2003 and revised in 2015 by the International Liaison Committee on Resuscitation (ILCOR): Drowning “is a process resulting in primary respiratory impairment from submersion/immersion in a liquid medium” [4].

During submersion (immersion of the body and the head), the respiratory stimulus cannot be suppressed for longer than about two minutes. Panic leads to aspiration of water and induces a laryngospasm as a protective reflex. Due to increasing hypoxia and hypercapnia, unconsciousness sets in, protective reflexes such as laryngospasm resolve consecutively and larger amounts of fluid (water, stomach contents) are aspirated [5]. Pulmonary atelectasis forms, which can lead to pulmonary edema and acute respiratory distress syndrome [1]. Consecutively bradycardia and circulatory arrest follow [6]. Drowning is therefore considered a death by suffocation (asphyxia) [7], which is why the elimination of hypoxia is the primary goal of medical treatment for drowning victims and the only way to restore spontaneous circulation [4,6,8,9].

Due to the exposure and depending on the submersion time, hypothermia coexists, which can affect all organ systems. Severe hypothermia with a body temperature of <28 °C can lead to cardiac arrhythmia and asystole. The initial respiratory acidosis increases through reduced metabolism and leads to hyperglycemia through reduced release of insulin and peripheral insulin resistance. In addition, the coagulation system can be disturbed. On the other hand, hypothermia may also have a neuroprotective effect, which is repeatedly cited as the cause of good clinical outcome after drowning incidents [5,10]. After rescue and restoration of cardiopulmonary function, secondary hypoxia-induced organ dysfunctions, disseminated intravascular coagulation, or possible rhabdomyolysis with development of acute kidney failure can be consecutive problems [5].

The consequences of hypoxia, hypercapnia, and acidosis caused by respiratory insufficiency, cardiac arrest, and hypothermia should be treated in drowning victims. As brain function is very vulnerable to hypoxia [9], the prognosis after a drowning incident depends mainly on the extent of hypoxic damage to the central nervous system [1,2,11].

For about 20 years, the main focus of research has been on defining prognostic parameters for drowning victims and developing action regimens based on those parameters [12]. Important factors discussed include submersion time, initial neurological and cardiopulmonary function, body temperature, laboratory parameters such as pH, glucose and lactate levels, type of prehospital care at the incident site, previous illnesses, age, gender, social background, and others [1,3,4,13,14,15,16,17,18,19,20,21,22].

Some survivors show severe neurological limitations [2] including minimal or vegetative states of consciousness [23]. Especially children with prolonged necessity of advanced life support (>30 min) have a very poor outcome [24]. The mortality rate in the first year after the incident is high [22,25]. As an anatomical correlate of hypoxic brain damage, magnetic resonance imaging revealed brain edema and structural changes, especially in the basal ganglia region, in all cases with poor outcome [24]. In addition to cognitive impairment, severe motor disorders such as spasticity and dystonia may also result [25,26].

A major goal of neurorehabilitation after drowning incidents is to restore functions and activities in order to facilitate psycho-motor development as well as prevent long-term damage caused by spasticity such as contractures, deformities, or scoliosis with the help of physical therapy, medication, operations, and aids [27,28,29]. Another component of rehabilitation is the treatment of comorbidities such as epilepsy, nutritional disorders, and osteoporosis. The patients and their relatives are cared for and guided by the multidisciplinary team, with the aim of achieving the best possible integration and participation of the patient in society [28,30].

The aim of this retrospective study is to describe incident characteristics and provide an overview of the patient journey of acute and intermediate care after drowning, with a special focus on rehabilitation and long-term outcome in the catchment area of the Zurich Children’s Hospital on the basis of medical records from the Zurich Children’s Hospital and the associated Swiss Children’s Rehab. In addition, prognostic parameters were used to classify new subcategories, evaluated and discussed on the basis of current literature.

## 2. Materials and Methods

### Study Design and Setting

This study was set up as a retrospective, single-center study. The clinical information system (CIS) of the University Children’s Hospital Zurich, Switzerland (CGM Phoenix^®^) was used to identify patients after drowning who were aged between 0–18 years at the time of the incident from 2000 to 2015 (catchment area approx. 1.2 million inhabitants). The search terms were as follows: drowning, near-drowning, drowning incident, drowning accident.

The cohort of drowning victims was admitted to either the ER, the intensive care unit directly via the trauma room and treated at the University Children’s Hospital of Zurich and/or the Swiss Children’s Rehab. The latter is a stand-alone pediatric rehabilitation clinic and part of the University Children’s Hospital Zurich, using the same CIS. After discharge from inpatient rehab, patients were regularly followed-up in the outpatient-rehab department and again documented using the same CIS.

All patient data were obtained from medical documentation stored in the CIS including emergency reports, hospital course reports, operation protocols, and radiology reports. The following variables were extracted for each case: Age at time of incident, incident date, swimming lessons, comorbidities. Incident: type of water body, supervision, submersion time, body temperature, resuscitation duration, bystander/lay resuscitation, neurological status (Glasgow Coma Scale (GCS)), laboratory values (e.g., pH, lactate, glucose). Patient journey (ER, pediatric intensive care unit (PICU), rehabilitation), Rehabilitation: duration of rehabilitation, functional tests such as the functional independence measure for children (weeFIM) [31], dependency on care, aids, functional scores such as Gross Motor Function Classification System (GMFCS), Manual Ability Classification System (MACS), Communication Function Classification System (CFCS), Eating and Drinking Ability Classification System (EDACS) [32]. These four classification systems, originally developed for children with cerebral palsy, allow a more precise description of the designated functions ranging from practically no limitation (level 1) to the most severe limitation of mobility, hand use, communication skills, and safety when eating (level 5) [32]. We also collected data regarding comorbidities, medications, interventions, additional therapies (physical therapy, occupational therapy, therapeutic education).

The main outcome classification was based on the Pediatric Cerebral Performance Category Scale (PCPCS) according to Fiser [33]. A good to medium outcome corresponds to a score of 1–3 points, patients with a PCPCS of 4–6 points are assumed to have severe neurological damage [34].

The outcome was defined as the status of consciousness according to the PCPCS. Selected parameters were recategorized into two subcategories on the basis of the current literature and tested for significant differences in outcome using the Mann–Whitney U Test. We hypothesized that a poor outcome is significantly more frequent in the subcategory with the more severe values of the parameters.

IBM’s Statistical Package for the Social Sciences (SPSS) was used for all statistical analyses (case number survey, descriptive statistics, Mann–Whitney U Test). If a *p* value was less than 0.05, a significant difference was assumed. If a *p* value was between 0.05 and 0.1, we classified this as a chance.

Patients whose data were explicitly not released for research purposes (marked in the clinical information system) were excluded from the study. Study approval was obtained from the local ethics committee of Zurich (KEK-ZH-Nr. 2014-0681).

## 3. Results

### 3.1. Study Population

Data sets from a total of 80 children and adolescents were used for the evaluation. Three patients identified had to be excluded from analysis as their record was not released for research. The male gender showed a higher number of cases with a significantly worse outcome (64% (*n* = 51), *p* value <0.001). The mean age was 4.79 years (SD ±3.7 years), the median was 3.86 years. The youngest patient was two months old, the oldest shortly before his 16th birthday (min. 0.18 years, max. 15.93 years). In four cases, a dysmorphic syndrome or a developmental delay and in one patient epilepsy was described as an underlying disease. One drowning incident occurred under the influence of alcohol, and drug use was not mentioned in any case.

### 3.2. Incident Characteristics

The majority of incidents, as shown in Figure 1, occurred in public swimming pools (*n* = 43, 54%). In 81.3% of the cases (*n* = 65), the children were unattended at the time of the incident. The responsibility for the victims was in 73.8% (*n* = 59) with a family member, mostly with the child’s parents. Only three drowning incidents (3.8%) occurred during swimming lessons with instructors and showed a good outcome in the PCPCS. 

### 3.3. Prehospital Care

Prehospital care includes care at the scene of incident or in the ambulance but not in the ER at the University Children’s Hospital. Twelve patients (9.6%) required oxygen administration via a nasal cannula or mask in case of respiratory insufficiency, in 21 cases (26%) intubation was necessary. 49 children (61%) needed cardiopulmonary resuscitation (CPR) after salvage, average resuscitation times ranged from a median time of 5 to 30 min. 

In 26 cases (32.5%), the CPR was initially performed by laypersons, in 21 cases (26%) trained first aiders were on site, 10 of them (12.5%) were lifeguards. CPR was successful in 63% (*n* = 31) of the cases. There is no reliable information available about whether the return of spontaneous circulation (ROSC) was achieved after arrival in the ER or at the scene. In all cases (*n* = 27, 34%) where initially no CPR was necessary, a good outcome was achieved. In 6 patients, no information about CPR was available.

### 3.4. Patient Journey

Figure 2 shows the patient journey after entering the hospital acute care. Four children were initially treated in other children’s hospitals in Switzerland and then referred to the Swiss Children’s Rehab for rehabilitation. In these four cases, the external patient reports for initial care and treatment in the emergency, intensive care, and normal wards were used and included in our study. In the case of five children, the further journey was not traceable on the basis of the medical records. Of the remaining 75 patients (94%) after a drowning incident, eight patients (10%) were discharged on the same day, 67 patients (84%) were indicated for hospital admission. Of these, 34 patients (43%) required initial treatment in the intensive care unit, seven patients (9%) died there. Moreover, 48 patients (60%) were discharged from the Children’s Hospital Zurich after their inpatient stay, and twelve children (15%) were transferred to the RCA for further care. Four (as of 25 March 2015) of these twelve children also passed away in the further course, in seven cases there was severe neurological restriction (PCPCS 4 + 5), one child was discharged from rehabilitation in good general condition (PCPCS = 1).

### 3.5. Analysis of Prognostic Parameters

We selected parameters that seem to be important according to the current scientific literature and our clinical experience and recategorized them in subcategories. We selected the first parameters glucose, lactate, and pH value that were obtained from the patients after admission to the ER. Then we tested whether these new artificial subcategories show differences in the outcome, measured with the PCPCS. Table 1 shows the case and percentage figures (*n*, %) of the respective parameter categories in relation to the outcome (PCPCS: good = 1–3, poor = 4–6). The Mann–Whitney U Test shows the significant differences between all categories considered and the outcome (see Table 1).

### 3.6. Rehabilitation Course and Long-Term Outcome

In the years 2000–2015, twelve patients after drowning incidents were treated as inpatients in the Swiss Children’s Rehab and after discharge followed-up in the rehab outpatient clinic. There were eight male and four female patients, the average duration of the first inpatient rehabilitation stay was almost one year (Ø 354 days). Four children died after their discharge from the children’s hospital (acute care): one child died two months after the incident, three patients died after 6–10 years. 

One of the twelve children was discharged from the inpatient rehabilitation in a neurologically normal state, according to PCPCS 1. Eleven of the twelve children were in a severely restricted state of consciousness (PCPCS 4 (*n* = 8) and PCPCS 5 (*n* = 3)) during the first rehabilitation stay and were completely dependent on care. In these children, most severe restrictions in all functional areas (GMFCS, MACS, CFCS, and EDACS) were found, correlating with the neurological condition. In all eleven cases, medication for spasticity therapy was administered, eight patients received additional injections for local reduction of the muscle tone and four patients received intrathecal baclofen therapy. In five cases, hip (sub)dislocation developed, in seven patients scoliosis occurred with need for bracing. Orthopedic surgeries were performed in six cases. In eleven cases of pronounced dysphagia, enteral nutrition was performed via percutaneous endoscopic gastrostomy. Six of the twelve patients suffered from epilepsy, with focal or generalized, single or recurrent seizures, or myoclonia, which were treated with long-term or intermittent anticonvulsive therapy. The patients required on average six medical aids and devices, including orthoses, standing aids, buggies and wheelchairs, braces, helmets, and others.

Long-term outcome: Seven of the twelve children were still seen in the outpatient rehabilitation department of the Children’s Hospital Zurich during data collection. The last documented consultation (outpatient or inpatient) was considered, which took place on average almost seven years after the drowning incident. Only one child showed improvement in the 3.7 years after the incident in terms of state of consciousness and functional independence index as well as in the CFCS and EDACS, of which each improved by one degree of severity. In the other six cases, the neurological health status and functional limitations (weeFIM, GMFCS, MACS, CFCS, and EDACS) were stable, one patient deteriorated increasingly during the inpatient rehabilitation stay and died in the course of the stay.

## 4. Discussion

Our study provides a first overview of the incident characteristics, acute treatment, rehabilitation, and long-term outcome of drowning victims in the catchment area of the Zurich Children’s Hospital, which is the biggest Children’s Hospital in Switzerland. In all, 80 patients were included in this descriptive study. The gender ratio was 64% for the male patients, most incidents occur between 0 and 7 years. The most frequent incident locations in our study population were public swimming pools. More than 80% of the children were unattended at the time of the incident. These results place emphasis on supervision of children, especially in the age of 0–5 years and male gender, next to public or private pools. We are aware, that this descriptive finding does not, however, lead to the general conclusion that public pools are particularly dangerous. No general epidemiological statements can be made with our study population. More children are likely to bathe in public pools than in open water or private pools. However, reduction of submersion time, as one of the most important prognostic factors, is closely interrelated with prevention measures such as surveillance (e.g., CCTV monitoring) and secure fencing of pools (public and private) and ponds to prevent unattended submersion in the first place. 

In four patients, a developmental delay or dysmorphic syndrome was described as a pre-existing condition. Consequently, it seems reasonable to intensify the supervision of children with pre-existing conditions near water. In the retrospective analysis, we were only able to collect limited sociodemographic data (gender and age). These are congruent with the literature [1,35,36]. We emphasize the importance to include such sociodemographic data in further investigations as they could be important factors to improve specific prevention measures. 

Treating hypoxia after a drowning incident is decisive for the clinical course [4,11]. Children after drowning incidents need supportive ventilation or cardiopulmonary resuscitation in more than 50% of cases [37]. In this study, 61% of the children needed CPR, of which 63% were successful. Effective CPR is crucial for the outcome [38,39], which is why further investment is to be made in improved prehospital care, in particular in the training of parents. At the time of admission, the children are assessed neurologically on the basis of the Glasgow Coma Scale (GCS) and respiratory parameters (respiratory rate, skin color, pulmonary edema, arterial blood gas analysis) in order to determine the further therapy regimen [40].

Damage to the central nervous system is the most important cause of mortality and morbidity after a drowning incident [1,2,11]. Based on the current literature, the initial neurological state (quantified by GCS) [41,42] and the submersion time [1,20,43] seem to be the most important prognostic parameters. However, both markers can only be used to a limited extent. Neurology is difficult to assess in partially hypothermic and sedated patients [5], and the data on submersion time is based on estimates of those present and are therefore inaccurate and incomplete [19], which is also reflected in the emergency reports that we have analyzed.

Medical decision-making in severe cases of drowning should consider prognostic parameters. In accordance with the literature and based on the available data (Table 1) and similar results of other authors, a poor outcome (PCPCS 4–6) can be assumed when the patient presents with the following:a GCS of 3 at time of admission [1,14,15,16,44],lack of motor response [3],no pupil reaction [1,15,16],a submersion time > 5 min [1,35,43,45,46],a body temperature < 30 °C [1,13,47],pH value < 7 [1,14],blood glucose ≥ 15 mmol/L [1],blood lactate ≥ 14 mmol/L [1],infiltrates in the chest X-ray [13,35].

From a clinical perspective, it seems reasonable not to consider a single factor but to analyze several values together and thus generate an estimate of outcome. Prospective studies with complete data sets are needed to build a robust model for prognosis. This is also stated in a recent systematic review regarding the prediction of outcome from scene factors known prior to rescue after drowning by Quan et al. [46].

In current literature [1,7,48], the outcome after a drowning incident is summarized as follows and is comparable with the results we obtained:75–80% of patients survive drowning without neurological sequelae,10% survive drowning with severe neurological deficits, and10–35% of drowning victims will die.

As our data also show, after a drowning incident, about 10% of the victims suffer from severe neurological deficits [7]. Some of the survivors remain in a vegetative or minimal conscious state (MCS) [29,49]. Our results of the PCPCS are consistent with this: Eleven of the twelve children with severe neurological deficits were in a vegetative state or MCS during the first rehabilitation stay, showed severe limitations in all functional areas, and were completely dependent on care. The prognosis of patients in a vegetative or MCS is of great medical and economic interest and well-studied [50,51]. If a child improves neurologically within the first 3 months after the incident, a good development is assumed [52]. However, if the vegetative state or MCS persists over nine months, restitution can only be expected in less than 5% of cases [50], which also corresponds to our data. Comparable to the results of Kriel et al. [52], where no patient improved cognitive or motor skills after anoxic brain damage over the years, in the present study only one patient in an MCS showed a slight improvement of cognitive ability. Life expectancy is reduced to two to five years and survival longer than 10 years is very unusual [53]. A recent population-based prospective cohort study (1974–1996) of Western Washington Drowning Registry (WWDR) investigating the long-term survival after drowning-related cardiac arrest in children and adults found that most cases of death occurred after termination of initial resuscitation or during the initial hospitalization [22]. The authors stated that the study could not estimate long-term neurologic outcomes beyond hospital discharge. 

We are aware of several limitations of this study. One limitation is the small number of patients especially in the category of severely injured children. Second, as this is a retrospective study, some data especially concerning the ROSC and the long-term outcome are missing. The large variability of the disease definitions and the assessment tools used in the literature make evaluation and comparison with other studies difficult. The GMFCS, CFCS, MACS, and EDACS scores are validated only for children with cerebral palsy. As there are no existing scores specifically for the description of functional abilities for children with chronic conditions after acquired brain injury, these are used in this study. We consider a bias in our long-term study group of drowned children referred to rehabilitation toward severe impairment, compared to the study describing neurocognitive outcome after drowning and resuscitation of by Suominen (2014) [54]. In 57% (*n* = 21) of their patients, they found long-term neurological deficits (median 8.1 years) and 40% had low full-scale intelligence quotients [54]. Due to the severity of the neurologic dysfunction of our study group, it was not possible to perform a neuropsychological examination for higher cortical functions such as IQ, memory, attention, executive verbal, visuospatial, or visuo-perceptual functions. In our descriptive study, we could not find reliable neurocognitive data of the less affected children from the medical records. In accordance with Vanagt [55], we would advocate to systematically schedule children with drowning for neuropsychological follow-up assessments and to make these data available for further studies as there is still missing knowledge. We are also aware that our data are not representative for epidemiology of drowning in Switzerland, as mild drowning cases will probably be admitted to the local ER in their community and not be referred to the Zurich University Children’s Hospital. Furthermore, sociocultural influences on drowning incidents have not been addressed in this retrospective analysis. To improve drowning prevention, further studies should investigate the impacts of sociocultural parameters and gender on drowning epidemiology and outcome. 

Further studies, especially prospective studies regarding the outcome, are needed in order to improve medical decision-making in severe cases of drowning. We suggest they should include submersion time, GCS at time of admission, body temperature at time of admission, blood pH, blood glucose, and blood lactate levels, as these parameters correlate with long-term outcome. The Utstein Style Guidelines for reporting drowning offer a framework for tracking outcome data including parameters of the different treatment periods (on site, ROSC, admission to hospital, and survival to hospital discharge, rehabilitation, and long-term neurological function), which would help facilitate future knowledge [4]. Improvement and persistent maintenance of awareness and prevention measures are crucial, as fatal outcomes occur and the long-time sequelae of drowning in surviving children and its consequences for their families are severe and are related to high healthcare costs.

## Figures and Tables

**Figure 1 children-07-00070-f001:**
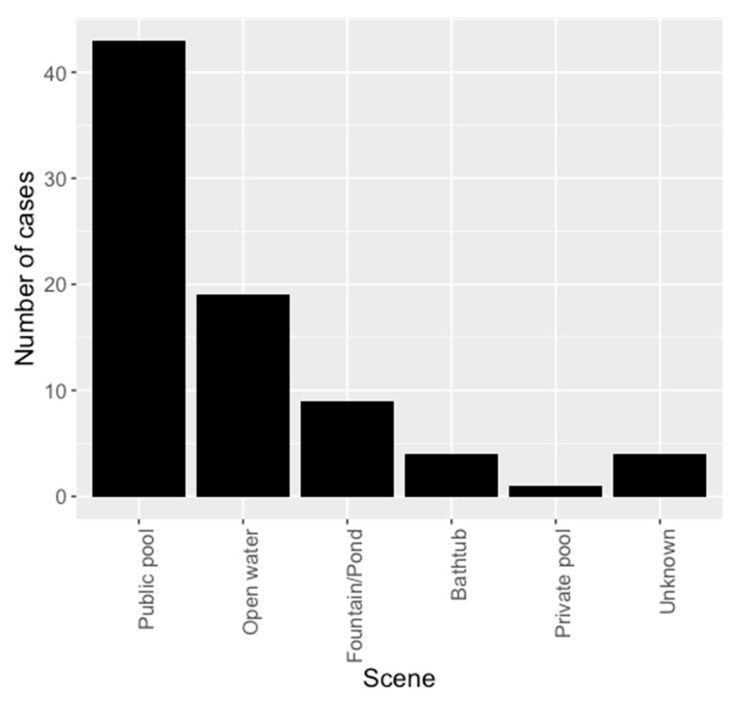
Incident characteristics describing frequency and place of incident.

**Figure 2 children-07-00070-f002:**
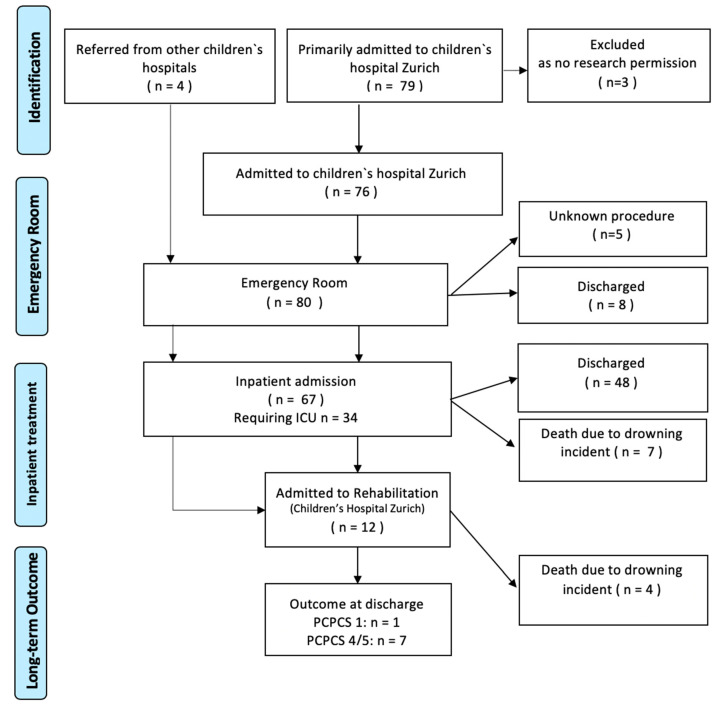
Patient journey (according to PRISMA flow diagram).

**Table 1 children-07-00070-t001:** New categorized parameters in correlation with outcome.

Parameter	Total Number (PCPCS 1–6) *n* = 80		Good Outcome (PCPCS 1–3) *n* = 62		Poor Outcome (PCPCS 4–6) *n* = 18		*u* Value, *p* Value (Mann–Whitney Test)
	*n*	%	*n*	%	*n*	%	
**Submersion time**							−4.561, <0.001
1 = time ≤5 min	35	43.75	33	94.3	2	5.7	
2 = time >5 min	12	15	3	25	9	75	
No data available	33	41.25	26	-	7	-	
**Glasgow Coma Scale (GCS) at time of admission**							−7.391, <0.001
1 = 4–15 points	59	73.75	56	94.9	3	5.1	
2 = 3 points	15	18.75	1	6.7	14	93.3	
No data available	6	7.5	5	-	1	-	
**Body temperature at time of admission**							−4.674, <0.001
1 = 32–37.5 °C	50	62.5	43	86	7	14	
2 ≤ 32 °C	10	12.5	1	10	9	90	
No data available	20	25	18	-	2	-	
**pH value**							−5.733, <0.001
1 ≤ 7	16	20	2	12.5	14	87.5	
2 ≥ 7	34	42.5	32	94.1	2	5.9	
No data available	30	37.5	28	-	2	-	
**Glucose**							−5.523, <0.001
1 ≤ 15 mmol/L	34	42.5	31	91.2	3	8.9	
2 ≥ 15 mmol/L	9	11.25	0	0	9	100	
No data available	37	46.25	31	-	6	-	
**Lactate**							−5.792, < 0.001
1 ≤ 14 mmol/L	36	45	34	94.4	2	5.6	
2 ≥ 14 mmol/L	15	18.75	2	13.3	13	86.6	
No data available	29	36.25	26	-	3	-	
**Chest X-Ray**							−3.371, 0.001
1 = normal	23	28.75	23	100	0	0	
2 = Infiltrate	33	41.25	20	60.6	13	39.4	
No data available	24	30	19	-	5	-	

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
