# Peer review of "Drowning in Children: Retrospective Analysis of Incident Characteristics, Predicting Parameters, and Long-Term Outcome"

_children, 2020, doi:10.3390/children7070070_

Round 1
Reviewer 1 Report
Thank you for the honor and pleasure to be invited to review the manuscript Drowning in children: retrospective analysis of risk factors, prognostic factors and long-term outcome. The authors have to be congratulated with this study that attempts to describe the journey of the patient from the accident site to discharge or rehabilitation and, in addition, to link this journey with prognostic factors. There is an increasing need to better understand when and how the prognosis of a drowned child can be predicted, while there is also a need to better understand the clinical course of non-fatal drowning.
At the same time, there are several remarks to be made regarding the methodology of the study, the presentation of the data and the structure and up-to-date content and references of the manuscript.
The study seems to be based on data coming from various data sources. It is not clear how the data have been collected and how accurate data collection and further processing has been. Some additional information is needed to understand which measures have been taken to obtain reliable and complete data. This is most of all true for data from the beginning of the journey (location, supervision, activity, bystander CPR) and the end of the journey (rehabilitation). If some data are less accurate, they better are not included. The introduction mentions that, in Switzerland, one out of 20 drowning is non-fatal. The number of admitted patients in this study is however much higher and also high compared to other studies (or the number discharged from the ER relatively low), which deserves more information about the inclusion criteria of the study; for example, can some children who were brought to the ER by their parents been missed in the study. If so, could this have affected the findings. The information in the statistics should be more specific and detailed regarding the tests for ordinal and continuous data; in the current version there seems no multiple regression analysis included, although this seems very useful.
The data regarding the patient journey (patient flow) should be more clearly presented as a PRISMA model, with special attention and focus to the group of patients that has been resuscitated and the neurological outcome in the various steps in the journey. Details that are still confusing or not sure from the descriptions are: did all patients who only received oxygen survived (and where did they receive oxygen: at the pool site by lifeguards, in the ambulance, in the ER), was the resuscitation successful at the scene or only after arrival in the ER; did all patients admitted to the ward survival. It seems that the neurological outcome is missing in 3 of the 11 patients that went to rehabilitation.
The analysis of the prognostic parameters needs more research rigor. In table 1, the cut-off points of the parameters need to be mentioned. It may make sense to calculate both the corrected and uncorrected p-values, while also a multiple regression analysis would be well in place here.
It is unclear on what evidence from this study the new set of prognostic parameters in table 2 has been selected; certainly not on the previous step in the study (multiple organ failure and chest x-ray were not mentioned in table 1). The selection seems to be most of all based on common sense and a subjective interpretation of the literature. It is also not clear at what time in the course of the journey the measures have been taken; is this the first measurement in the ED or, for example, the highest value during the first 24 hours? Multi-organ failure will most of the time develop after 3 days and not at admission; this is a different moment to establish the prognosis compared to parameters in the ED. There may also be significant bias: for example a chest X-ray will normally not been ordered in a child without any symptoms. The table nor the text make clear how the full set of parameters result in a formula for prognosis. Would it be possible to have an equally accurate prognostic instrument with less data. The concept that is explored in table 1 and 2 is interesting but should be based on a more robust methodology.
The study seems to have explored profoundly the neurologic outcome, The methodology of this part of the study and the presentation of the findings needs more information and structure.
The discussion mentions many neurological scores without further explanation of its value and weakness. Some more information may be needed, if this information is relevant to understand the data that emerge from this study.
Although the information in the introduction and discussion is interesting and adequate in itself, both introduction and discussion could match better to the research objective. Most information in the introduction is not directly related or focussed to the research objective. The introduction and discussion need also to become more up-to-date. Many references are 30-40 years old, while more recent studies have resulted in advancements of the understanding of the prognosis of drowned persons. Examples of these references include Reynolds 2019: 128; Venema 2018:19; Quan 2016:63; Kieboom 2015:350; Vanagt 2014: 98; Suominen 2014 and 2012:20.
Also some references seem not adequate. F.e. the source paper about the definition of drowning is by Van Beeck; 2005 and the Utstein style for drowning has been updated in 2017. At the same time, the reference list includes a number of interesting publications in the German language that are often not mentioned in the English literature.
Some details
Title: risk factors are not the objective of this study
Table 2: Regarding the temperature, the range 30-32C seems to be missing.
Reviewer 2 Report
This is a very useful paper adding to the literature on the long term outcomes and eventual fatality of non-fatal drowning, for which data is scarce.
Abstract
- Line 9 - Not all non-fatal drownings lead to disability, please rephrase
Methods
- Was patient journey data only collected for patients that attended the same hospital for ongoing care and rehabilitation? Or was patient data also found where care was sought elsewhere, and if so, how was it accessed? Please clarify in the methodology how this data was searched for.
- Please provide more detail on the timeframe followed for patient journey, and how longitudinal data was gathered – it is not clear how the finding in the abstract around four dying due to complications in the 10 years following the incident was found.
Discussion
- Please include some discussion on the Prognostic parameters that can be addressed through systematic solutions, e.g. reduction of submersion time, in order to improve non-fatal drowning outcomes.
- A second limitation may be that all non-fatal drowning events may not be brought to hospitals, especially if the child appears mostly ‘normal’, so the statistics on outcomes of drowning incidents may be an over-representation of the total non-fatal drowning events.
Reviewer 3 Report
Thank you for giving us (me and my doctoral student) the opportunity to review the manuscript, “Drowning in Children: Retrospective Analysis of Risk 3 Factors, Prognostic Factors and Long-Term Outcome”
We tend not to use the word “accident” in injury anymore. “Incident” is preferred.
There are frequent comma splices that need to be addressed throughout the text. Paragraphs are generally too short to be considered paragraphs. Please address paragraph length throughout the text.
Your abstract needs to be more specific in limiting the findings to Switzerland. For instance, “…is most important for prevention of drowning accidents in Switzerland.” Your results cannot be generalized to other places.
Language needs to be modified at times throughout the manuscript. For example, on page 3 line 136, the authors state that “The responsibility for the victims was in 73.8% (n=59) with a family member, mostly with the child parents”. This could be modified to “In 73.8% (n=59) of the cases, family members (mostly the child’s parents) were responsible for the accidents”, for example. There is also a need to check the punctuation and grammar of the manuscript.
Page 1, line 15, I think there should be some note on what age-range childhood falls between. It does not seem to be conceptually linked as a stage in the text. I think what would benefit the manuscript is clearly articulating in the abstract and method section what ages childhood ranges between. This is especially so given that the results at times indicate, such as on page 3 line 126-128, that “The male sex showed a higher number of cases with a significantly worse outcome (64% (n=51), p-value <0.001). The mean age was 4.79 years (STD +/- 3.7y)”. A standard deviation of +/- 3.7 years can be a lot in injury prevention. For children, there is a large difference between a 1 year old and an 8 and a half year old. I think this should be spoken to more in the discussion.
Page 1, line 36, what do the authors mean by “natural” and “non-natural” deaths. I think this needs some specificity. Is this terminology synonymous with preventable and non-preventable injuries that lead to fatalities? If so, I think preventable should be used instead, as natural is much more open to interpretation and designates that there are normal ways to die.
Page 1, line 40, you say “… the consequences of which are serious for the victims”. I think this relates back to my previous point of the necessity for specificity or example. What is serious, for example? Serious in injury prevention can be very broad and can vary depending on the population’s interpretation of it. For example, while parents may find serious to be debilitating or long-lasting consequence, children may find it to be a severe headache. I think you need more specificity and more references to show that the terminology relates to pre-established categories for the term. For example, on page 2 line 73, you use “severe neurological limitations” and then describe what these could be. This helps guide the reader to what “severe” includes. The same is necessary when identifying what is serious.
On p. 2 there are two paragraphs that are only a sentence long. They should be revised.
- 2 “the patient and his relatives” – please use gender inclusive language.
- 3 Who was conducting the swimming lessons? A parent or another person (instructor)?
On page 3, line 116-117, you state that “Selected parameters were recategorised into two subcategories on the basis of the current literature and tested for significant differences in outcome”. This could be expanded upon to clearly state what the two subcategories are in relation to one or two sentences on how they relate back to the established literature.
On page 3 line 123, you state that “If a p-value was between 0.05 and 0.1, we classified this as a tendency”. I’m curious as to why this was and I think that your rationale should be explained in the manuscript. The word choice may be more appropriately selected as “chance” and not “tendency”. If this is used because of the sample size, then I think this needs to be referenced. As this relates to the results and discussion of your results, it should be backed by a reference indicating that in this field this terminology is the best to use. I also think that if this relates to your sample size calculation, then perhaps a reference to this would strengthen the claim. For example, see Nahm’s (2017) thoughts on this. Nahm, F. S. (2017). What the P values really tell us. The Korean Journal of Pain, 30(4), 241.
- 6 Please provide the sociodemographic data.
On page 6, line 206 - 209 you state that “The gender ratio is 64% for the male patients, most accidents occur between 0-7 years. Most frequent accident location are public and private swimming pools. More than 80% of the patients were unattended at the time of the accident. These results place emphasis on supervision of children, especially of the age 0-5 years, next to public or private pools”. I believe your statement may be strengthened by including a gendered recommendation. Thus, based on your findings, I think it would be important to say something along the lines of, “These results place emphasis on the necessity for the supervision of children, especially boys between the age 0-5 years, next to public or private pools”.
- 7 You call for “improved primary care,” especially knowledge of CPR in parents. Can parents really be considered “primary care providers”? Maybe there is a language difference, but in North America, a primary care provider would be a family doctor or a nurse – not a parent.
- 8 line 270 The semi-colon should be a comma.
On page 8 line 275 when you list the weaknesses of the study, I believe the manuscript would benefit from a few sentences listing the limitation of not including a discussion on the sociocultural implications of your findings. For example, in risk research, boys typically take more and greater risks than girls in many environments. It may thus benefit the discussion to link this important finding back to the need to understand gender in relation to drowning prevention, especially for boys aged 0-5 years old, as your results showcase. Your results can provide insight into the necessity to understand gender in relation to this outcome, due to the serious consequences associated with submersion.
